# GOI: Find 3D Gaussians of Interest with an Optimizable Open-vocabulary Semantic-space Hyperplane

Yansong Qu[*]
quyans@stu.xmu.edu.cn
Xiamen, Fujian, China

Shaohui Dai[*]
daish@stu.xmu.edu.cn
Xiamen, Fujian, China

Xinyang Li
imlixinyang@gmail.com
Xiamen, Fujian, China

Jianghang Lin
hunterjlin007@stu.xmu.edu.cn
Xiamen, Fujian, China

Liujuan Cao[†]
caoliujuan@xmu.edu.cn
Xiamen, Fujian, China

Shengchuan Zhang
zsc_2016@xmu.edu.cn
Xiamen, Fujian, China

Rongrong Ji
rrj@xmu.edu.cn
Xiamen, Fujian, China
Key Laboratory of Multimedia
Trusted Perception and Efficient
Computing, Ministry of Education of
China, Xiamen University, China

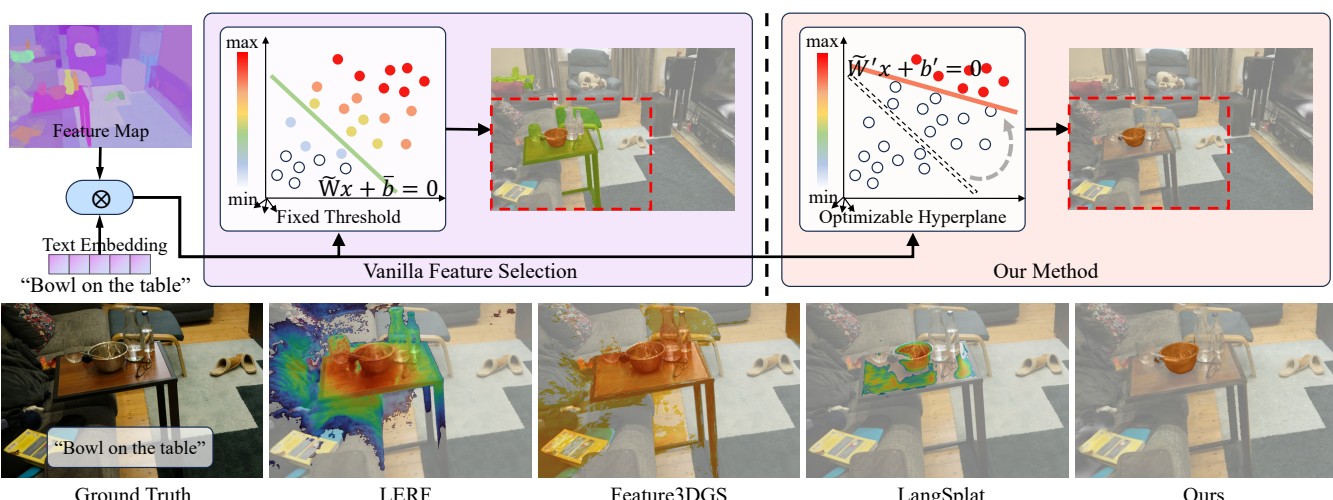

**Figure 1: We propose GOI, an innovative approach to 3D open-vocabulary scene understanding based on 3D Gaussian Splatting [20]. In the top row, we emphasize our key contribution: the Optimizable Semantic-space Hyperplane (OSH). Instead of relying on a manually set, fixed empirical threshold for relative feature selection, which frequently lacks universal accuracy, OSH is fine-tuned for each query to accurately locate target regions in response to natural language prompts. The bottom row showcases our superior performance in open-vocabulary querying compared to other approaches.**

[*]Equal Contribution.
[†]Corresponding author.

## Abstract

3D open-vocabulary scene understanding, crucial for advancing augmented reality and robotic applications, involves interpreting and locating specific regions within a 3D space as directed by natural language instructions. To this end, we introduce GOI, a framework that integrates semantic features from 2D vision-language foundation models into 3D Gaussian Splatting (3DGS) and identifies 3D Gaussians of Interest using an Optimizable Semantic-space Hyperplane. Our approach includes an efficient compression method that utilizes scene priors to condense noisy high-dimensional semantic features into compact low-dimensional vectors, which are

subsequently embedded in 3DGS. During the open-vocabulary querying process, we adopt a distinct approach compared to existing methods, which depend on a manually set fixed empirical threshold to select regions based on their semantic feature distance to the query text embedding. This traditional approach often lacks universal accuracy, leading to challenges in precisely identifying specific target areas. Instead, our method treats the feature selection process as a hyperplane division within the feature space, retaining only those features that are highly relevant to the query. We leverage off-the-shelf 2D Referring Expression Segmentation (RES) models to fine-tune the semantic-space hyperplane, enabling a more precise distinction between target regions and others. This fine-tuning substantially improves the accuracy of open-vocabulary queries, ensuring the precise localization of pertinent 3D Gaussians. Extensive experiments demonstrate GOI's superiority over previous state-of-the-art methods. The dataset, model, and code are available at https://quyans.github.io/GOI-Hyperplane/.

## CCS Concepts

• **Computing methodologies → Scene understanding**.

## Keywords

Open-vocabulary, 3D scene understanding, 3D Gaussian Splatting, Semantic Field, Hyperplane

**ACM Reference Format:**
Yansong Qu, Shaohui Dai, Xinyang Li, Jianghang Lin, Liujuan Cao, Shengchuan Zhang, and Rongrong Ji. 2024. GOI: Find 3D Gaussians of Interest with an Optimizable Open-vocabulary Semantic-space Hyperplane. In *Proceedings of the 32nd ACM International Conference on Multimedia (MM '24), October 28-November 1, 2024, Melbourne, VIC, Australia.* ACM, New York, NY, USA, 10 pages. https://doi.org/10.1145/3664647.3680852

## 1 Introduction

The field of computer vision has witnessed a remarkable evolution in recent years, driven by advancements in artificial intelligence and deep learning. A critical aspect of this progress is the enhanced ability of computer systems to interpret and interact with the three-dimensional world. The growing complexity in technology use has spurred a significant demand for advanced 3D visual understanding. This evolution brings to the fore the significance of the open-vocabulary querying task [5, 28, 34] — the capacity to process and respond to user queries formulated in natural language, enabling a more natural and flexible interaction between users and the digital world. Such advancements hold the potential to enhance how human navigate and manipulate complex three-dimensional data [14, 19, 47], bridging the gap between human cognitive abilities and computerized processing [7, 21].

Due to the scarcity of large-scale and diverse 3D scene datasets with language annotations, earlier Methods [21, 29] distill the multimodal knowledge from off-the-shelf vision-language models, such as CLIP [42] and LSeg [24], into Neural Radiance Fields (NeRF) [35]. However, NeRF's implicit representation limits its speed and accuracy, hindering practical application. Recently, the 3D Gaussian Splatting (3DGS) [20] has emerged as an effective representation of 3D scenes, with explorations in constructing semantic fields [40, 49, 60]. This semantics lifting approach requires pixel-aligned

features, whereas CLIP encodes a image into one global semantic feature. [21, 30, 49] utilize a multi-scale feature pyramid that incorporates CLIP embeddings from image crops. However, this approach results in blurred semantic boundaries, even with DINO [3] features, leading to unsatisfactory query results.

In this work, we introduce 3D **G**aussians **O**f **I**nterest (GOI). We utilize the vision-language foundation model APE [48] to extract pixel-aligned semantic features from multi-view images. GOI leverages these semantic features to reconstruct a 3D Gaussian semantic field. Due to the explicit nature of 3DGS, embedding high-dimensional features into each 3D Gaussian is computationally demanding. To address this, we introduce the Trainable Feature Clustering Codebook (TFCC), which compresses noisy high-dimensional features using scene priors. This approach significantly reduces storage and rendering costs while maintaining the informational capacity of each feature. Moreover, current open-vocabulary query strategies call for setting a fixed empirical threshold to ascertain features proximate to the query text. This, however, results in a failure to precisely query the targets. We introduce the Optimizable Semantic-space Hyperplane (OSH) to address this issue. OSH is fine-tuned by the Referring Expression Segmentation (RES) model, which aims to identify binary segmentation masks in 2D RGB images for text queries and is recognized for its robust spatial and localization capabilities. The OSH enhances GOI's spatial perception for more precise phrasal queries like "the table under the bowl", aligning query results more closely with target regions. Additionally, we have meticulously expanded and annotated a subset of the Mip-NeRF360 [1] dataset, tailored for the open-vocabulary query task. Owing to our method's proficient 3D open-vocabulary scene understanding, it is practical for a range of downstream applications, notably scene manipulation and editing.

In summary, the main contributions of our work include:

- We propose GOI, an innovative framework based on 3D Gaussian Splatting for accurate 3D open-vocabulary semantic perception. The Trainable Feature Clustering Codebook (TFCC) is further introduced to efficiently condense noisy high-dimensional semantic features into compact, low-dimensional vectors, ensuring well-defined segmentation boundaries.

- We introduce the Optimizable Semantic-space Hyperplane (OSH), which eschews the fixed empirical threshold for relative feature selection due to its limited generalizability. Instead, OSH is fine-tuned for each text query with the off-the-shelf RES model to precisely locate target regions.

- Extensive experiments demonstrate that our method outperforms the state-of-the-art methods, achieving substantial improvements in mean Intersection over Union (mIoU) of 30% on the Mip-NeRF360 dataset [1] and 12% on the Replica dataset [50].

## 2 Related Work

### 2.1 Neural Scene Representation

Recent methods in representing 3D scenes with neural networks have made substantial progress. Notably, Neural Radiance Fields (NeRF) [35] have excelled in novel view synthesis, producing highly realistic new viewpoints. However, NeRF's reliance on a neural

network for complete implicit representation of scenes leads to tedious training and rendering times. Many subsequent methods [6, 12, 18, 36, 43, 44] have concentrated on improving its performance. In order to enhance the quality of surface reconstruction, [10, 13, 33, 52, 53] uses the signed distance function (SDF) for surface expression and uses a novel volume rendering scheme to learn an SDF representation. On the other hand, some approaches [8, 9, 39, 41, 55, 56] have explored the combination of implicit and explicit representations, utilizing traditional geometric structures, such as point clouds or mesh, to enhance NeRF's performance and to enable more downstream tasks. Kerbl et al. proposed 3D Gaussian Splatting (3DGS) [20], which greatly accelerates the rendering speed of novel view synthesis and achieves high-quality scene reconstruction. Unlike NeRF that represents a 3D scene implicitly with neural networks, 3DGS represent a scene as a set of 3D Gaussian ellipsoids, and accomplish efficient rendering by rasterizing the Gaussian ellipsoids into images. The technique adopted by 3DGS, which entails encoding scene information into a collection of Gaussian ellipsoids, provides distinct advantages [25, 26, 54]. It permits easy manipulation of specific parts in the reconstructed scene without significantly affecting other components. We have extended the 3DGS to achieve open-vocabulary 3D scene perception.

## 2.2  2D Visual Foundation Models

Foundation Models (FM) are becoming an impactful paradigm in the content of AI. They are typically pre-trained on vast amounts of data, possess numerous model parameters, and can be adapted to a wide range of downstream tasks [2]. The efficacy of 2D visual foundation models is evident in multiple visual tasks, such as object localization [31] and image segmentation [15–17]. The incorporation of multimodality substantially amplifies the perceptual ability of these models. For instance, CLIP [42], by using contrastive learning, aligns the outputs of text encoders and image encoders into the unified feature space. Similarly, SAM [22] showcases immersive capabilities as a promptable segmentation model, delivering competitive, even superior zero-shot performance vis-à-vis earlier fully-supervised models. DINO [4, 37], a self-supervised Vision Transformer (ViT) model, is trained on vast unlabeled images. The model deciphers a semantic representation of images, encompassing components such as object boundaries and scene layouts.

Moreover, recent efforts are focused on leveraging existing pretrained models, thereby pushing the limit of Foundation Models. Grounding DINO [32] represents an open-set object detector executing target detection based on textual descriptions. It utilizes CLIP and DINO as basic encoders, and proposes a tight fusion approach for better synthesizing of visual-language information. Grounded SAM [45] integrates Grounding DINO with SAM, facilitating the detection and segmentation for arbitrary queries. APE [48] is a universal visual perception model designed for diverse tasks like segmentation and grounding. Rigorously designed visual-language fusion and alignment modules enable APE to detect anything in an image swiftly without heavy cross-modal interactions.

## 2.3  3D Scene Understanding

Earlier works, such as Semantic NeRF [59] and Panoptic NeRF [11], introduced the transfer of 2D semantic or panoptic labels into 3D radiance fields for zero-shot scene comprehension. Following this, [23, 51] capitalized on pixel-aligned image semantic features, which they lifted to 3D, rather than relying on pre-defined semantic labels. Vision-language models like CLIP exhibited impressive performance in zero-shot image understanding tasks. A subsequent body of work [21, 23, 30] proposed leveraging CLIP and CLIP-based visual encoders to extract dense semantic features from images, with the aim of integrating them into NeRF scenes.

The recently proposed 3D Gaussian Splatting has achieved leading benchmarks in areas of novel view synthesis and reconstruction speed. This advancement has made the integration of 3D scenes with feature fields more efficient. LangSplat [40], LEGaussians [49], Feature 3DGS [60], Gaussian Grouping [57] explored the integration of pixel-aligned feature vectors from 2D models like LSeg, CLIP, DINO and SAM into 3D Gaussian frameworks so as to enabling 3D open-vocabulary query and localization of scene areas.

## 3  Methods

## 3.1  Problem Definition and Method Overview

Given a set of posed images $I = \{I_1, I_2, \ldots, I_K\}$, a 3D Gaussian scene $S$ can be reconstructed using the standard 3D Gaussian Splatting technique [20] based on $I$. Our method expands $S$ with open-vocabulary semantics, enabling us to precisely locate the Gaussians of interest based on a natural language query.

We begin by recapping the vanilla 3D Gaussian Splatting (Sec. 3.2). Figure 2 illustrates the overview pipeline of our method. Initially, we utilize a frozen image encoder, well-aligned with the language space, to process each image $I_k$ and derive the 2D semantic feature maps $V = \{V_1, V_2, \ldots, V_K\}$ (Sec. 3.3). To integrate these 2D high-dimensional feature maps into 3DGS, while ensuring minimal storage and optimal computational performance, Trainable Feature Clustering Codebook (TFCC) is proposed (Sec. 3.4). We expand 3DGS to reconstruct 3D Gaussian Semantic Field (Sec. 3.5). Following this, we explain how to utilize the RES model to optimize the Semantic-space Hyperplane, thereby achieving accurate open-ended language queries in 3D Gaussians (Sec. 3.6).

## 3.2  Vanilla 3D Gaussian Splatting

3D Gaussian Splatting utilizes a set of 3D Gaussians, essentially Gaussian ellipsoids, which bears a significant resemblance to point clouds, to model the scene and accomplish fast rendering by efficiently rasterizing Gaussians into images, given cameras poses. Specifically, each 3D Gaussian is parameterized by its centroid $x \in \mathbb{R}^3$, a 3D anisotropic covariance matrix $\Sigma$ in world coordinates, an opacity value $\alpha$, and spherical harmonics (SH) $c$. In the rendering process, 3D Gaussians are projected on to the 2D image plane, which transforms 3D Gaussian ellipsoids into 2D ellipses. $\Sigma$ is transformed to $\Sigma'$ in camera coordinates:

$$\Sigma' = JW\Sigma W^T J^T, \tag{1}$$

where $W$ denotes the world-to-camera tranformation matrix and $J$ is the Jacobian matrix for the projective transformation. In practical, $\Sigma$ is decomposed into a rotation matrix $R$ and a scaling matrix $S$:

$$\Sigma = RSS^T R^T. \tag{2}$$

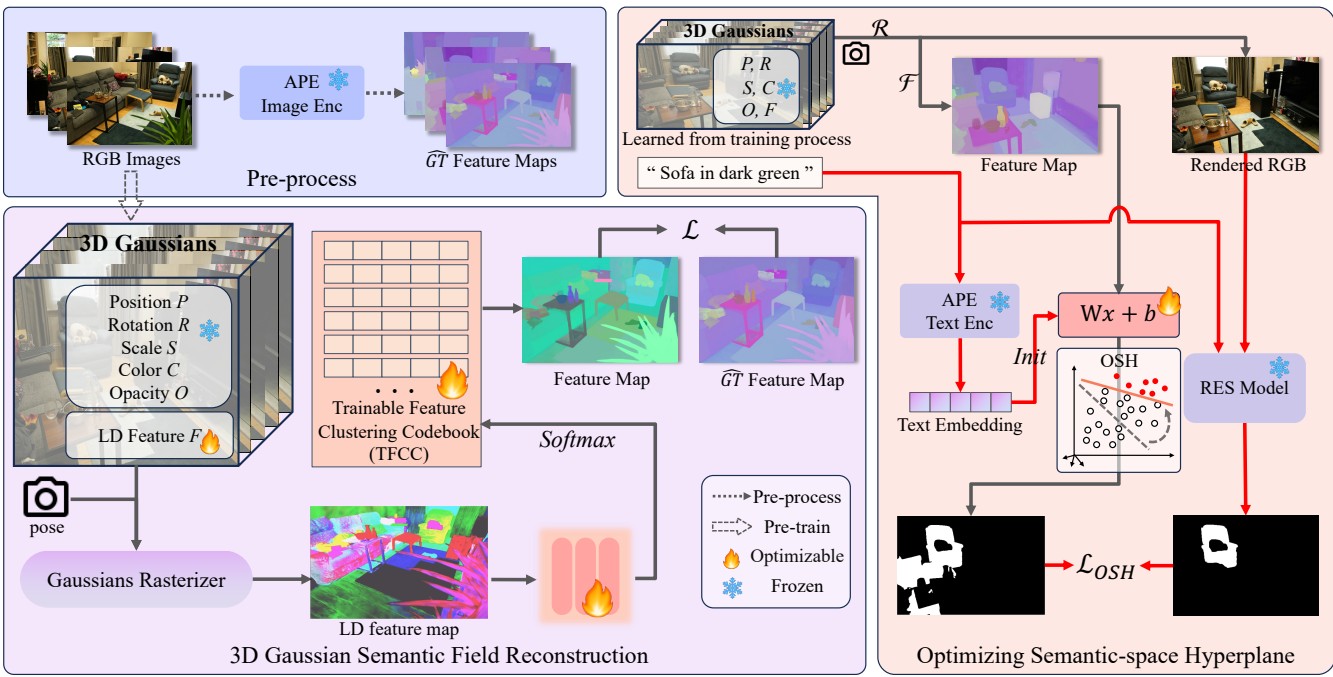

**Figure 2: The framework of our GOI. Top left: Reconstruction of a 3D Gaussian scene [20], encoding multi-view images. Bottom left: The optimization process. For each training view, a low-dimensional (LD) feature map is rendered through Gaussian Rasterizer and transformed into a predicted feature map via the Trainable Feature Clustering Codebook (TFCC). Right: The pipeline illustrates open-vocabulary querying. The processes denoted by $\mathcal{R}$ and $\mathcal{F}$ correspond to rendering and feature map prediction, respectively. The red line indicates operations exclusive to the initial query with a new text prompt. During these operations, the Optimizable Semantic-space Hyperplane (OSH) is fine-tuned to more precisely delineate the target region.**

This decomposition is to ensure that $\Sigma$ is physically meaningful during the optimization. To summarize, the learnable parameters of the $i$-th 3D Gaussian are represented by $\theta_i = \{x_i, R_i, S_i, \alpha_i, c_i\}$.

A volumetric rendering process, similar to NeRF, is then employed in the rasterization to compute the color $C$ of each pixel.

$$C = \sum_{i \in G} c_i \alpha_i T_i, \qquad (3)$$

where $G$ denotes a set of 3D Gaussians sorted by their depth, and $T_i$ represents the transmittance, defined as the cumulative product of the opacity values of Gaussians that superimpose on the same pixel, computed through $T_i = \prod_{j=1}^{i-1}(1 - \alpha_j)$.

### 3.3 Pixel-level Semantic Feature Extraction

Prior research has broadly employed CLIP for feature lifting in the 3D radiance field, owing to its superior capability in managing open-vocabulary queries. [23, 60] use LSeg [24] to extract pixel-aligned CLIP features. However, LSeg proves inadequate in recognizing long-tail objects. To compensate for CLIP's limitation for yielding only image-level features, methodologies such as [21, 40, 49] adopt a feature pyramid approach, using cropped image encoding to represent local features. These methods extract pixel-level features from the CLIP model, but the encoded feature maps lack geometric correspondence to the scene objects. As such, pixel-aligned DINO features are introduced and predicted simultaneously with the CLIP

features, thus bounding CLIP with the object geometry. Leveraging the success of SAM, [27, 40] utilizes SAM explicitly to constrain the object-level boundaries of the features. However, using multiple models for feature extraction substantially increases the complexity for training and image prepocessing.

We leverage the Aligning and Prompting Everything All at Once model (APE) [48], which is also able to align the features of vision and language. In APE, a fixed language model formulates language features, and a visual encoder is trained from scratch. The core of the visual encoder, derived from the DeformableDETR [61], provides APE with formidable detection and localization capacities. Additionally, APE possesses efficient modules for vision-language fusion and alignment. These modules diminish cross-modal interaction and thus reducing computational costs. Therefore, APE presents a robust solution for feature lifting. For this purpose, we make minor modifications to the APE model to extract pixel-aligned features with fine boundaries efficiently (~2s per image). We treat the encoded pixel-aligned feature maps as the pseudo ground truth features, denoted as $\widehat{GT}$.

We extract APE feature maps from all training viewpoints and embed them into each 3D Gaussian to reconstruct a 3D semantic field. During the open-vocabulary querying process, we use the pretrained APE language model to encode the language prompts.

## 3.4 Trainable Feature Clustering Codebook

Due to APE being trained on mass data and the need to align text and image features, it results in a higher feature dimensionality (256). As the previous works [40, 49] have mentioned, directly lifting high-dimensional semantic features into each 3D Gaussian results in excessive storage and computational demands. The semantics of a single scene cover only a small portion of the original CLIP feature space. Therefore, leveraging scene priors for compression can effectively reduce these costs. On the other hand, due to the inherent multi-view inconsistency of encoded 2D semantic feature map, Gaussians tend to overfit each training viewpoint, inheriting this inconsistency and causing discrepancies between 3D and 2D within an object. Therefore, we introduce the Trainable Feature Clustering Codebook (TFCC), which leverages scene priors to compress the semantic space of a scene into an $N$ length codebook. This approach effectively reduces redundant and noisy semantic features while preserving sufficient scene information and clear semantic boundaries.

## 3.5 3D Gaussian Semantic Fields

We introduce a low-dimensional semantic feature, symbolized as $f$, into each 3D Gaussian, capitalizing on the redundancy of high-dimensional semantics across the scene and dimensions to facilitate efficient rendering. To create a 2D semantic representation, we employ a volumetric rendering process similar to color rendering (Sec. 3.2) onto the low-dimensional semantic feature.

$$\hat{f} = \sum_{i \in G} f_i \alpha_i T_i. \tag{4}$$

$\hat{f}$ is the pixel-wise low-dimensional feature. We utilize an MLP as a feature decoder to obtain logits $e$, which are subsequently activated by the Softmax function to find the corresponding TFCC entry's index. This process acquires the feature $v$ in the high-dimensional semantic space for each $\hat{f}$. Given that volumetric rendering is essentially a process of weighted averages, the 3D Gaussian feature $f$ and the rendered 2D pixel-wise feature $\hat{f}$ are fundamentally equivalent. The low-dimensional feature $\hat{f}$ and $f$ can both be recovered to semantic feature $v$ through the MLP decoder $\mathcal{D}$ and the TFCC $\mathcal{T}$ with $N$ entries,

$$v = \mathcal{T}\left[\operatorname*{argmax}_{j=1,2,...,N}(e_j)\right], \tag{5}$$

where $e = \mathcal{D}(\hat{f})$ and $e \in \mathbb{R}^N$. Thus, both 2D and 3D features can be restrained to a compact and finite semantic space.

Initially in the semantic field optimization, we focus on learning the TFCC from $\widehat{GT}$ features. To enhance reconstruction efficiency, we adopt $k$-means clustering through $\widehat{GT}$ feature maps $V$ for the codebook initialization. Also, we find some resemblance between the learning of TFCC and the contrastive pre-training from CLIP: Features in the codebook are to align with the $\widehat{GT}$ features, and each $\widehat{GT}$ feature, denoted as $v_{gt}$, is assigned to one TFCC entry with the highest similarity. However, the assignment of a pixel feature to a particular entry is not predetermined, rather it pivots on similarity. Therefore, we devise a self-supervised loss function

aimed at reducing the self-entropy of the clustering process.

$$\mathcal{L}_{ent} = -\sum_{j=1}^N p_j \log(p_j), \tag{6}$$

where $p_j = \text{Softmax}\left(\cos\langle v_{gt}, \mathcal{T}[j]\rangle \cdot q\right)$ and $q$ is the annealing temperature. To accelerate the process, we also optimize the entry with the highest similarity, introducing a loss similar to [49],

$$d = \operatorname*{argmax}_{j=1,2,...,N}\left(\cos\langle v_{gt}, \mathcal{T}[j]\rangle\right), \tag{7}$$

$$\mathcal{L}_{max} = 1 - \cos\langle v_{gt}, \mathcal{T}[d]\rangle. \tag{8}$$

Thus, the loss in optimizing the TFCC is

$$\mathcal{L}_T = \lambda_{ent}\mathcal{L}_{ent} + \lambda_{max}\mathcal{L}_{max}. \tag{9}$$

Subsequently, we undertake a joint optimization of the low-dimensional features $\hat{f}$ and the MLP decoder $\mathcal{D}$. Ideally, the feature recovered from low-dimensional feature should closely correlate with the $\widehat{GT}$ feature $v_{gt}$. As a result, we impose a stronger constraint geared towards aligning the entries' logits of the low-dimensional features with the assigned $\widehat{GT}$ entry $d$,

$$\mathcal{L}_{joint} = \|e - \text{onehot}(d)\|_2^2. \tag{10}$$

Finally, to bolster the robustness of this procedure, we introduce an end-to-end regularization, directly optimizing the cosine similarity of 2D semantic feature and corresponding ground truth,

$$\mathcal{L}_{e2e} = 1 - \cos\langle v_{gt}, v\rangle. \tag{11}$$

The comprehensive loss function designated for our semantic field reconstruction process is represented as $\mathcal{L}$,

$$\mathcal{L} = \mathcal{L}_T + \lambda_{joint}\mathcal{L}_{joint} + \lambda_{e2e}\mathcal{L}_{e2e}. \tag{12}$$

## 3.6 Optimizable Semantic-space Hyperplane

Thanks to the vision-language models like CLIP and APE, which align features well in image and text spaces. Our 3D Gaussian semantic field, once trained, supports open-vocabulary 3D queries with any text prompt. Most existing methods enable open-vocabulary queries by computing the cosine similarity between semantic and text features, defined as follows: $\cos(\theta) = \frac{\phi_{img} \cdot \phi_{text}}{\|\phi_{img}\|\|\phi_{text}\|}$, where $\phi_{img}$ and $\phi_{text}$ represent the image and text features, respectively. After normalizing the features, the score can be simplified as $Score = \phi_{img} \cdot \phi_{text}$. The higher the score, the greater the similarity between the two features. By manually setting an empirical threshold $\tau$, regions with score exceeding $\tau$ are retained, thus enabling open-vocabulary queries. The aforementioned process can be conceptualized as a hyperplane separating semantic features into two categories: features of interest and features not of interest, based on the queried text feature and $\tau$. The hyperplane is represented as follows:

$$\widetilde{W}x + \bar{b} = 0. \tag{13}$$

Here $\widetilde{W}$ denotes the queried text feature, $x$ represents semantic features and $\bar{b}$ is the bias derived from $\tau$. However, the empirical parameter $\tau$ is not universally applicable to all queries, often resulting in an inability to precisely locate target areas. Consequently, we propose the Optimizable Semantic-space Hyperplane (OSH). Utilizing a RES model, such as Grounded-SAM [45], we obtain a 2D binary mask of the target area and optimize the hyperplane via one-shot logistic regression. This optimization ensures that the

classification results of the hyperplane more closely align with the target area of the query.

As shown on the right side of Figure 2, From a specific camera pose, an RGB image and a feature map are obtained through the rgb and semantic feature rendering processes described in Sec. 3.5, respectively. For a text query $t$, the text encoder of APE generates a text embedding $\phi_{text}$, which is used as the initial weight of the hyperplane $Wx + b = 0$. The Feature Map is classified by the hyperplane, resulting in the prediction of a binary mask $m$. The text query $t$ and the RGB image are processed by the RES Model to generate a binary mask $\hat{m}$ of the target area as the pseudo-label. This mask is subsequently used with $m$ in logistic regression to optimize $W$ and $b$. We fine-tune the OSH with the objective:

$$\mathcal{L}_{OSH} = -\frac{1}{P} \sum_{i=1}^{P} [w \cdot \hat{m}_i \log(\sigma(m_i)) + (1 - \hat{m}_i) \log(1 - \sigma(m_i))], \quad (14)$$

where $P$ denotes all samples, $\sigma(\cdot)$ denotes Sigmoid function. Following the one-shot logistic regression, the optimized Semantic-space Hyperplane can be represented by

$$\widetilde{W}'x + b' = 0. \quad (15)$$

Note that the parameters of the 3D Gaussians remain frozen during this process. The red lines in Figure 2 indicate operations that occur only during the initial query with a new text prompt. Subsequently, the OSH can delineate regions of interest in both 2D rendered views and 3D Gaussians. Specifically, for a semantic feature $F$, derived from a 2D semantic feature map at pixel $p$ or from a 3D Gaussian $g$, if $\widetilde{W}'F + b' > 0$, it indicates that $F$ is sufficiently close to the queried text, warranting retention of $p$ or $g$ in the query results set.

## 4 Implementation Details

Our method is implemented based on 3D Gaussian Splatting [20]. We modified the CUDA kernel to render semantic features on the 3D Gaussians, ensuring that the extended semantic features also support gradient backpropagation. Our model, based on a 3D Gaussian Scene reconstructed via vanilla 3D Gaussian Splatting, can be trained on a single A100-40G GPU in approximately 10 minutes.

## 5 Experiments

### 5.1 Evaluation Setup

**Datasets.** To assess the effectiveness of our approach, we conduct experiments on two datasets: The Mip-NeRF360 dataset [1] and the Replica dataset [50]. Mip-NeRF360 is a high-quality real-world dataset that contains a number of objects with rich details. It is extensively used in 3D reconstruction. We selected four scenes (Room, Bonsai, Garden, and Kitchen), both indoors and outdoors, for our evaluations. Additionally, we designed an open-vocabulary semantic segmentation test set under these scenes. We manually annotated a few prominent objects in each scene, providing 2D masks and descriptive phrases, such as "sofa in dark green". Replica is a 3D synthetic dataset that features high-fidelity indoor scenes. Each scene comprises RGB images along with semantic annotations. We conducted experiments on four commonly used scenes from the Replica dataset: office0, office1, room0, and room1. For a given viewpoint image, our evaluation concentrates on assessing the effectiveness of single-query results within an open-vocabulary

context rather than obtaining a similarity map for all vocabularies in a closed set and deciding mask regions based on similarity scores [27, 30, 60]. Therefore, in designing our experiments, we drew inspiration from the methodologies of refCOCO and refCOCOg [58]. For each semantic ground truth in the Replica test set, we cataloged the class names present and sequentially used these class names as text queries to quantitatively measure the performance metrics.

**Baseline Methods and Evaluation Metrics.** To assess the accuracy of open-vocabulary querying results, we employ mean Intersection over Union (mIoU), mean Pixel Accuracy (mPA), and mean Precision (mP) as evaluation metrics. Additionally, to evaluate model performance metrics, we measure the training duration and the rendering time.

### 5.2 Comparisons

We conduct a comparative evaluation of our approach in contrast with LangSplat [40], Gaussian Grouping [57], Feature 3DGS [60], and LERF [21].

**Qualitative Results.** We present the qualitative results produced by our method alongside comparisons with other approaches. Figure 3 offers a detailed showcase of the open-vocabulary query performance on the Mip-NeRF360 test data. It especially highlights the utilization of phrases that describe the appearance, texture, and relative positioning of different objects.

LERF [21] generates imprecise and vague 3D features, which hinder the clear discernment of boundaries between the target region and others. Feature 3DGS [60] employs a 2D semantic segmentation model LSeg [24] as its feature extractor. However, like LSeg, it lacks proficiency in handling open-vocabulary queries. It frequently queries all objects related to the prompt. Gaussian Grouping [57] leverages the instance mask via SAM [22] to group 3D Gaussians into 3D instances devoid of semantic information. It uses Grounding DINO [32] to pinpoint regions of interest for enabling 3D open-vocabulary queries. However, this approach leads to granularity issues, often identifying only a fraction of the queried object, such as the major part of "green grass" or the flower stem from the "flowerpot on the table". LangSplat [40] uses SAM to generate object segmentation masks and subsequently employs CLIP to encode these regions. However, this strategy results in CLIP encoding only object-level features, leading to an inadequate understanding of the correlations among objects within a scene. For instance, when querying "the tablemat next to the red gloves", it erroneously highlights the "red gloves" rather than the intended "tablemat".

Our methodology effectively uses semantic redundancy to cluster features into a TFCC, enabling efficient encoding of diverse object features. Consequently, this approach precisely pinpoints objects such as the sofa, grass, and road while maintaining accurate boundaries. Our strategy further excels at discerning the intricate interrelationships among various objects within a scene. Unlike LangSplat, we encode entire images with the image encoder to integrate scene-level information into the semantic features. Additionally, we deploy dynamically optimize a semantic-space hyperplane, effectively filtering out unnecessary objects from the 3D Gaussians of Interest. For instance, in the cases of "flowerpot on the table" and "the tablemat next to the red gloves", we successfully segment the primary subjects of the phrase rather than the secondary objects.

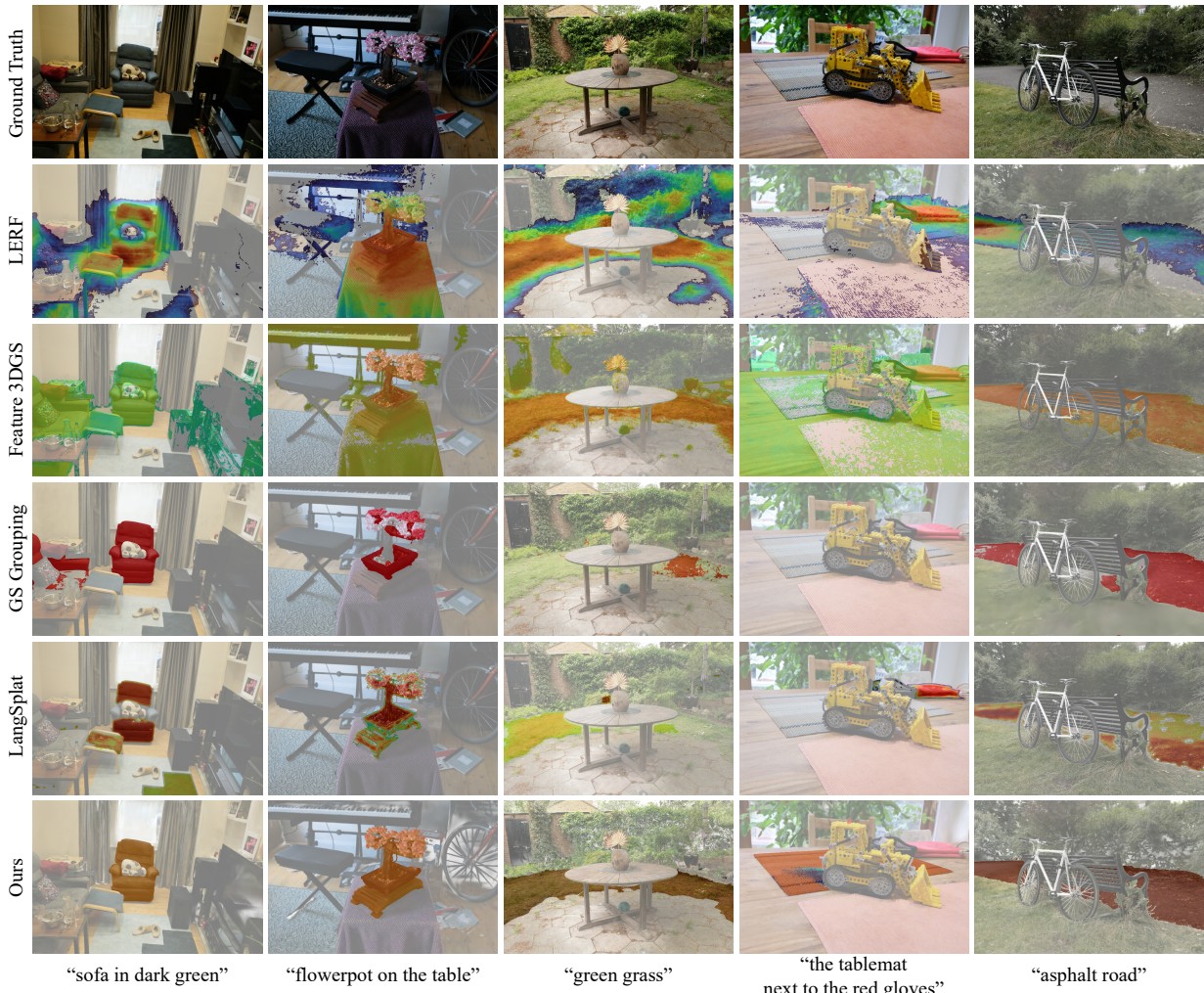

Figure 3: Visualization comparisons of open-vocabulary querying results are presented. From top to bottom: Ground truth, querying results from LERF [21], Feature 3DGS [60], Gaussian Grouping [57], LangSplat [40], and our method. From left to right, the images display the querying results corresponding to text descriptions, which are noted at the bottom line.

Table 1: Evaluation metrics for comparing our method with others on Mip-NeRF360 [1] evaluation dataset.

| Method | mIoU | mPA | mP |
|---|---|---|---|
| LERF [21] | 0.2698 | 0.8183 | 0.6553 |
| Feature 3DGS [60] | 0.3889 | 0.8279 | 0.7085 |
| GS Grouping [57] | 0.4410 | 0.7586 | 0.7611 |
| LangSplat [40] | 0.5545 | 0.8071 | 0.8600 |
| Ours | **0.8646** | **0.9569** | **0.9362** |

Table 2: Evaluation metrics for comparing our method with others on Replica [50] evaluation dataset.

| Method | mIoU | mPA | mP |
|---|---|---|---|
| LERF [21] | 0.2815 | 0.7071 | 0.6602 |
| Feature 3DGS [60] | 0.4480 | 0.7901 | 0.7310 |
| GS Grouping [57] | 0.4170 | 0.73699 | 0.7276 |
| LangSplat [40] | 0.4703 | 0.7694 | 0.7604 |
| Ours | **0.6169** | **0.8367** | **0.8088** |

**Quantitative Results.** Table 1 and Table 2 provide a comparative analysis of the efficacy of our work relative to other projects across multiple datasets. As displayed, our segmentation precision significantly exceeds existing open-vocabulary methods, including LERF and recent 3DGS-based approaches. We observed a substantial mean Intersection over Union (mIoU) improvement of 30% on the Mip-NeRF360 dataset and 12% on the Replica dataset, respectively.

Moreover, Table 3 underscores the effectiveness of our approach. We detail the image pre-processing time for extracting semantic

**Table 3: Time evaluation for training and rendering on Mip-NeRF360 [1] dataset.**

| Method | Pre-process | Training | Total | FPS |
|---|---|---|---|---|
| LERF [21] | **3min** | 40min | **43min** | 0.17 |
| Feature 3DGS [60] | 25min | 10h 23min | 10h 48min | ~10 |
| GS Grouping [57] | 27min | 113min | 140min | **~100** |
| LangSplat [40] | 50min | 25+99min | 174min | ~30 |
| Ours | 8min | **25+12min** | 45min | ~30 |

**Table 4: Evaluation metrics for ablation studies on Mip-NeRF360 [1] dataset.**

| Setting | mIoU | mPA | mP |
|---|---|---|---|
| Baseline | 0.4753 | 0.8638 | 0.7577 |
| w/o OSH | 0.6282 | 0.9464 | 0.8157 |
| w/o TFCC | 0.7537 | 0.9011 | 0.9115 |
| Full model | **0.8646** | **0.9569** | **0.9362** |

features, scene reconstruction duration, total training time, and rendering frame rates for each approach under consideration. Using a highly efficient visual encoder derived from APE, we reduced single image encoding time to 2 seconds. Furthermore, unlike the training strategies of LERF, Feature 3DGS, and Gaussian Grouping, which start training from scratch, both LangSplat and our method build 3D semantic fields from pre-trained 3DGS scenes. To ensure fairness, the time required for pre-training scenes using 3D Gaussian Splatting (25 minutes) is included in the overall training time calculation of ours and LangSplat. Through meticulous TFCC design and training regularization, we successfully reconstruct a semantic field in under 12 minutes.

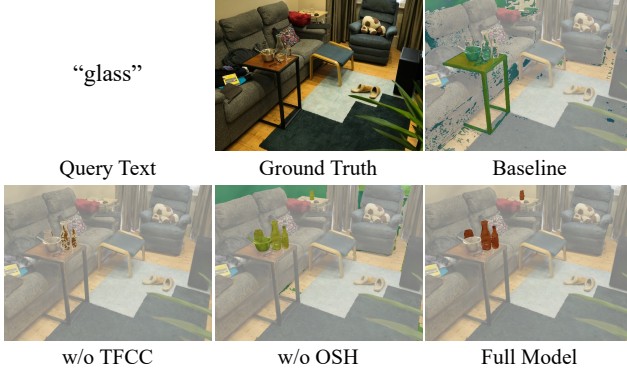

"glass"

Query Text · Ground Truth · Baseline
w/o TFCC · w/o OSH · Full Model

**Figure 4: Visualization comparison of ablation experiments using the query text "glass".**

## 5.3 Ablation Studies

To discover each component's contribution to 3D open-vocabulary scene understanding in our proposed pipeline, a series of ablation experiments are conducted for the Mip-NeRF360 dataset [1], using the same 2D semantic feature maps extracted from the APE [48]

image encoder. We employ the approach of lifting compressed low-dimensionality semantic features into 3D Gaussians as our baseline. This is contrasted with results from models not utilizing the TFCC module, those not employing the OSH module, and the results from the complete model.

As illstrated in Table 4, OSH and TFCC are critical to the effectiveness of our approach; without them, there would be a significant deterioration in performance (-27% ~ -12% mIoU). As shown in Figure 4, the baseline model (top-right) struggles due to its scattered features, making it difficult for the model with the OSH module (bottom-middle) to identify a suitable hyperplane. In contrast, the model with TFCC (bottom-left) demonstrates more clustered features and distinct semantic boundaries.

## 5.4 Application

Our method can be applied to a variety of downstream tasks, with the most direct application being the editing of 3D scenes. As shown in the figure 5, we use the text query "flowerpot on the table" to locate the 3D Gaussians of interest. Our method enables the highlighting of target areas, localized deletion, and movement. Furthermore, by integrating with Stable-Diffusion [46], We can employ the Score Distillation Sampling (SDS) [38] loss function to achieve high-quality 3D generation tasks in specific areas.

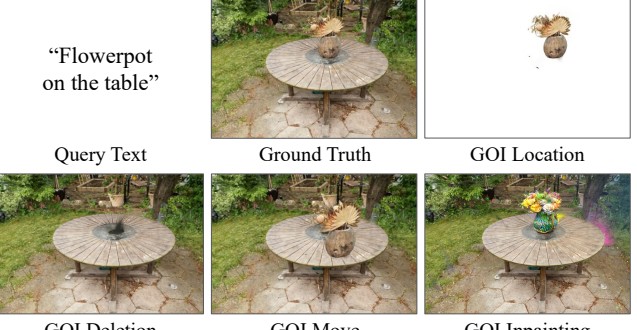

"Flowerpot on the table"

Query Text · Ground Truth · GOI Location
GOI Deletion · GOI Move · GOI Inpainting

**Figure 5: Visualization of scene manipulation results using our method. The query text is used to locate the 3D Gaussians of interest (GOI). "A beautiful vase" is used as the prompt for the 3D inpainting process after locating the GOI.**

## 6 Conclusion

In this paper, we introduce GOI, a method for reconstructing 3D semantic fields, capable of delivering precise results in 3D open-vocabulary querying. By leveraging the Trainable Feature Clustering Codebook, GOI effectively compresses high-dimensional semantic features and integrates these lower-dimensional features into 3DGS, significantly reducing memory and rendering costs while preserving distinct semantic feature boundaries. Moreover, moving away from traditional methods reliant on fixed empirical thresholds, our approach employs an Optimizable Semantic-space Hyperplane for feature selection, thereby enhancing querying accuracy. Through extensive experiments, GOI has demonstrated improved performance over existing methods, underscoring its potential for downstream tasks, such as localized scene editing.

## Acknowledgments

This work was supported by National Science and Technology Major Project (No. 2022ZD0118201), the National Science Fund for Distinguished Young Scholars (No.62025603), the National Natural Science Foundation of China (No. U21B2037, No. U22B2051, No. U23A20383, No. 62176222, No. 62176223, No. 62176226, No. 62072386, No. 62072387, No. 62072389, No. 62002305 and No. 62272401), and the Natural Science Foundation of Fujian Province of China (No.2022J06001).

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
