# OpenReview forum: "GOI: Find 3D Gaussians of Interest with an Optimizable Open-vocabulary Semantic-space Hyperplane"
_acmmm.org/ACMMM/2024/Conference — MM2024 Poster_

### Official Review · Reviewer_DVrW · 2024-05-22

**Rating:** 4
**Confidence:** 4

**Summary:**

The paper introduces a novel framework for 3D open-vocabulary scene understanding. The method leverages semantic features extracted from 2D vision-language models, integrating them into a 3D Gaussian splatting architecture. This integration facilitates the identification and localization of 3D Gaussians of Interest (GOI) based on natural language prompts. The paper claims substantial improvements over existing methods, particularly in terms of accuracy and the ability to handle open-vocabulary queries.

**Strengths:**

+ Innovative Approach: The combination of 3D Gaussian splatting with semantic features from 2D models is innovative. This approach allows for more precise localization and understanding of 3D scenes from natural language inputs.
&nbsp;
+ Optimizable Semantic-space Hyperplane (OSH): The introduction of OSH as a method to fine-tune the feature selection process dynamically in response to natural language queries is a significant advancement. It addresses the common issue of fixed thresholds that cannot adapt to various queries.
&nbsp;
+ Extensive Evaluation: The paper presents a thorough experimental setup, including comparisons with several state-of-the-art methods. The use of multiple datasets and extensive metrics (mIoU, mPA, mP) to evaluate their method adds credibility to their claims.
&nbsp;
+ Practical Applications: The paper does well to discuss potential practical applications, particularly in augmented reality and robotic navigation, which are areas of significant interest within the community.

**Limitations:**

- Complexity and Computation Requirements: The method appears to be computationally intensive, involving multiple components such as feature extraction, 3D Gaussian splatting, and the optimization of a semantic-space hyperplane. The practicality of deploying such a system in real-time applications may be limited.
&nbsp;
- Dependency on Training Data: The performance of the system heavily relies on the quality and diversity of the annotated 3D scene datasets. The paper does not address how the system would perform with less curated or noisy data.
&nbsp;
- Generalization Concerns: While the results are promising, the generalization of the model to a wider array of scenes and languages (beyond the tested datasets and English language prompts) remains unclear.
&nbsp;
- Hyperparameter Sensitivity: The method involves several layers of optimization and feature selection, which may be sensitive to hyperparameter settings. The paper does not discuss the sensitivity analysis or the robustness of the method against variations in hyperparameters.

**Suitability:**

3

---

### Official Review · Reviewer_qyGh · 2024-05-22

**Rating:** 2
**Confidence:** 3

**Summary:**

The paper proposes a feature distillation method using a learnable codebook inside 3D gaussian splatting for open vocabulary querying in 3D scenes. The paper also introduces Optimizable Semantic-space Hyperplane (OSH), which do away with the need for a manual threshold  in pixel localization. OSH can be finetuned for each text query and is more generalizable .

**Strengths:**

The learnable codebook usage inside gaussian spplatting for avoiding noisy features is a novel component. Optimizable Semantic-space Hyperplane (OSH) proposed in the work do away with the need for a manual threshold  in pixel localization.

**Limitations:**

Distilling knowledge from vision-language Models and constructing  semantic  fields inside  3D Gaussians has been explored in a number of previous works.  The paper mentions one of its contributions as having a lifting approach with  pixel aligned semantic fields . Section 3.5 in the paper discusses how 3D Gaussian Semantic field is obtained by volumetric rendering process similar to color rendering.  However this part is fundamentally similar to the 3D Gaussian semantic field creation discussed in Feature 3DGS [1].

The major difference from [1] is that the authors have used a Trainable Feature Clustering Codebook citing  ‘redundant and noisy high dimensional semantic features’ and ‘computational needs’ . How is the size, N,  of codebook fixed ?  What is the impact of varying ‘N’ ?  Also there is no ablation study presented to showcase the impact on accuracy and computation if the codebook is not used in the existing dataset (Replica).

Additionally the impact of OSH module is showcased only on Mip-NeRF360 annotations which I assume the authors have contributed. Ablation studies of OSH and TFCC on Replica  is also needed to establish the proposed method’s usefulness.

Table 3 showcase the training time advantage of the proposed method. The major difference in timing is attributed to the fact the paper starting with a pre-trained model of 3D gaussians. It is not clear if this can be adopted by other methods as well or is there any special design considerations that is specific to this paper that allows such an initialization and subsequent saving of training time.

[1] Feature 3DGS: Supercharging 3D Gaussian Splatting to Enable Distilled Feature Fields

Minor Comments:
Line 283 -  we utilize an frozen image encoder  we utilize a frozen image encoder

**Suitability:**

3

---

### Official Review · Reviewer_ZQBs · 2024-05-25

**Rating:** 4
**Confidence:** 2

**Summary:**

This paper focuses on 3D open-vocabulary semantic perception and introduces GOI, an innovative framework based on 3D Gaussian Splatting for accurate semantic analysis. GOI integrates semantic features from 2D vision-language foundation models into 3D Gaussian Splatting (3DGS), and it identifies 3D Gaussians of Interest using an Optimizable Semantic-space Hyperplane. This manuscript also includes an efficient compression method that utilizes scene priors to condense noisy, high-dimensional semantic features into compact, low-dimensional vectors. These vectors are then embedded in 3DGS. Additionally, the framework enhances feature management through the Trainable Feature Clustering Codebook (TFCC), which organizes and optimizes the feature space for improved processing efficiency and accuracy.

**Strengths:**

This paper presents a method that achieves comparable results on the benchmark Mip-NeRF360 Replica. It conceptualizes the feature selection process as a division of the feature space via hyperplanes, retaining only those features that are highly relevant to the query. This approach overcomes the limitations of traditional methods, which often lack universal accuracy and face challenges in precisely identifying specific target areas.

**Limitations:**

I am not an expert in this area, so I will refer to the opinions of other reviewers to determine my final rating.
I have the following questions:
It would be beneficial to see results on ScanNet200.
Could you also provide outcomes when directly using CLIP features?

**Suitability:**

2

---

### Meta-Review · Area_Chair_9cW3 · 2024-07-02

**Recommendation:** Accept (Poster)
**Confidence:** 5

**Metareview:**

The reviewers all recommend accepting this paper because of the design novelty and sufficient experimental validation. AC also agrees with the reviewers and recommends an acceptance of the submission.